# GANMEX: One-vs-One Attributions using GAN-based Model Explainability

## Abstract

Attribution methods have been shown as promising approaches for identifying key features that led to learned model predictions. While most existing attribution methods rely on a baseline input for performing feature perturbations, limited research has been conducted to address the baseline selection issues. Poor choices of baselines limit the ability of one-vs-one explanations for multi-class classifiers, which means the attribution methods were not able to explain why an input belongs to its original class but not the other specified target class. Achieving one-vs-one explanation is crucial when certain classes are more similar than others, e.g. two bird types among multiple animals, by focusing on key differentiating features rather than shared features across classes. In this paper, we present GANMEX, a novel approach applying Generative Adversarial Networks (GAN) by incorporating the to-be-explained classifier as part of the adversarial networks. Our approach effectively selects the baseline as the closest realistic sample belong to the target class, which allows attribution methods to provide true one-vs-one explanations. We showed that GANMEX baselines improved the saliency maps and led to stronger performance on perturbation-based evaluation metrics over the existing baselines. Existing attribution results are known for being insensitive to model randomization, and we demonstrated that GANMEX baselines led to better outcome under the cascading randomization of the model.

## 1 Introduction

Modern Deep Neural Network (DNN) designs have been advancing the state-of-the-art performance of numerous machine learning tasks with the help of increasing model complexities, which at the same time reduces model transparency. The need for explainable decision is crucial for earning trust of decision makers, required for regulatory purposes Goodman & Flaxman (2017), and extremely useful for development and maintainability.

Due to this, various attribution methods were developed to explain the DNNs decisions by attributing an importance weight to each input feature. In high level, most attribution methods, such as integrated gradient (IG) (Sundararajan et al. (2017)), DeepSHAP (Lundberg & Lee (2017)), DeepLift (Shrikumar et al. (2017)) and Occlusion (Zeiler & Fergus (2013)), alter the features between the original values and the values of some baseline instance, and accordingly highlight the features that impacts the model's decision. While extensive research has been conducted on the attribution algorithms, research regarding the selection of baselines is rather limited, and it is typically treated as an afterthought. Most existing methodologies by default apply a uniform-value baseline, which can dramatically impact the validity of the feature attributions (Sturmfels et al. (2020)), and as a result, existing attribution methods showed rather unperturbed output even after complete randomization of the DNN (Adebayo et al. (2018)).

In a multi-class classification setting, existing baseline choices do not allow specifying a target class, and this has limited the ability for providing a class-targeted or one-vs-one explanation, meaning explaining why the input belongs to class A and not a specific class B. These explanations are crucial when certain classes are more similar than others, as often happens for example when the classes have a hierarchy among them. For example, in a classification task of apples, oranges and bananas, a model decision for apples vs oranges should be based on their color rather than the shape

since both an apple and orange are round. This would intuitively only happen when asking for an explanation of 'why apple and not orange' rather than 'why apple'.

In this paper, we present GAN-based Model EXplainability (GANMEX), a novel methodology for generating one-vs-one explanations leveraging GAN. In a nutshell, we use GANs to produce a baseline image which is a realistic instance from a target class that resembles the original instance. A naive use of GANs can be problematic because the explanation generated would not be specific to the to-be-explained DNN. We lay out a well-tuned recipe that avoids these problems by incorporating the classifier as a static part of the adversarial networks and adding a similarity loss function for guiding the generator. We showed in the ablation study that both swapping in the DNN and adding the similarity loss are critical for resulting the correct explanations. To the best of our knowledge, GANMEX is the first to apply GAN for explaining DNN decisions, and furthermore the first to provide a realistic baseline image, rather than an ad-hoc null instance.

We showed that GANMEX baselines can be used with a variety of attribution methods, including IG, DeepLIFT, DeepSHAP and Occlusion, to produce one-vs-one attribution superior compared with existing approaches. GANMEX outperformed the existing baseline choices on perturbation-based evaluation metrics and showed more desirable behavior under the sanity checks of randomizing DNNs. Other than its obvious advantage for one-vs-one explanations, we show that by replacing only the baselines and without changing the attribution algorithms, GANMEX greatly improves the saliency maps for binary classifiers, where one-vs-one and one-vs-all are equivalent.

## 2    RELATED WORKS

### 2.1    ATTRIBUTION METHODS AND SALIENCY MAPS

Attribution methods and their visual form, saliency maps, have been commonly used for explaining DNNs. Given an input $x = [x_1, ..., x_N] \in \mathbb{R}^N$ and model output $S(x) = [S_1(x), ..., S_C(x)] \in \mathbb{R}^C$, an attribution method for output $i$ assign contribution to each pixel $A_{S,c} = [a_1, ..., a_N]$. There are two major attribution method families: Local attribution methods that are based on infinitesimal feature perturbations, such as gradient saliency (Simonyan et al. (2014)) and gradient*input (Shrikumar et al. (2016)), and global attribution methods that are based on feature perturbation with respect to a baseline input (Ancona et al. (2018)). We focus on global attribution methods since they tackle the gradient discontinuity issue in local attribution methods, and they are known to be more effective on explaining the marginal effect of a feature's existence (Ancona et al. (2018)). In this paper, we discussed five popular global attribution methods below:

**Integrated Gradient (IG)** (Sundararajan et al. (2017)) calculates a path integral of the model gradient from a baseline image $\tilde{x}$ to the input image $x$: $\mathcal{IG}_i = (x_i - \tilde{x}_i) \int_{\alpha=0}^{1} \partial_{x_i} S(\tilde{x} + \alpha(x - \tilde{x})) d\alpha$. The baseline is commonly chosen to be the zero input and the integration path is selected as the straight path between the baseline and the input.

**DeepLIFT** (Shrikumar et al. (2017)) addressed the discontinuity issue by performing backpropagation and assigns a score $C_{\Delta x_i \Delta t}$ to each neuron in the networks based on the input difference to the baseline $\Delta x_i = x_i - \tilde{x}_i$ and the difference in the activation to that of the baseline $\Delta t = t(x) - t(\tilde{x})$, that satisfies the summation-to-delta property $\sum_i C_{\Delta x_i \Delta t} = \Delta t$.

**Occlusion** (Zeiler & Fergus (2013); Ancona et al. (2018)) applies full-feature perturbations by removing each feature and calculating the impacts on the DNN output. The feature removal was performed by replacing its value with zero, meaning an all zero input was implicitly used as the baseline.

**DeepSHAP** (Chen et al. (2019); Lundberg & Lee (2017); Shrikumar et al. (2017)) was built upon the framework of DeepLIFT but connecting the multipliers of attribution rule (rescale rule) to SHAP values, which are computed by 'erasing features'. The operation of erasing one or more features require the notion of a background, which is defined by either a distribution (e.g. uniform distribution over the training set) or single baseline instance. For practical reasons, it is common to choose a single baseline instance to avoid having to store the entire training set in memory.

**Expected Gradient** (Erion et al. (2019)) is a variant of IG that calculates the expected attribution over a prior distribution of baseline input, usually approximated by the training set $X_T$, meaning

$\mathcal{EG}_i = \mathbb{E}_{\tilde{x} \sim X_T \alpha \sim U(0,1)}(x - \tilde{x})_i \partial_{x_i} S(\tilde{x} + \alpha(x - \tilde{x}))$ where $U$ is the uniform distribution. In other words, the baseline of IG is replaced with a uniform distribution over the samples in the training set.

A crucial property of the above methods is their need for a baseline, which is either explicitly or implicitly defined. In what follows we show that these methods are greatly improved by modifying their baseline to that chosen by GANMEX.

## 2.2 THE BASELINE SELECTION PROBLEM

Limited research has been done on the problem of baseline selection so far. A simple "most natural input", such as zero values of all numerical features is commonly chosen as the baseline. For image inputs, uniform images with all pixels set to the max/min/medium values are commonly chosen. The static baselines frequently cause the attribution to only focus on or even overly highlight the area where the feature values are different from the baseline values, and hide the feature importance where the input values are close to the baseline values (Sundararajan & Taly (2018); Adebayo et al. (2018); Kindermans et al. (2017); Sturmfels et al. (2020)).

Several none-static baselines have been proposed in the past, but each of them suffered from its own downsides (Sturmfels et al. (2020)). Fong & Vedaldi (2017) used blurred images as baselines, but the results are biased toward highlighting high-frequency information from the input. Bach et al. (2015) make use of the training samples by finding the training example belonging to the target class closest to the input in Euclidean distance. Even though the concept of minimum distance is highly desirable, but in practice, the nearest neighbor selection in high dimensional space can frequently lead to poor outcome, and most of the nearest neighbors are rather distant from the original input.

Along the same concept, expected gradient simply samples over all training instances instead of identifying the closest instance (Erion et al. (2019)). Expected gradient benefits from ensembling in a way similar to that of SmoothGrad, which averages over multiple saliency maps produced by imposing Gaussian noise on the original image (Smilkov et al. (2017); Hooker et al. (2019)). We claim however that averaging over the training set does not solve the issue; for example, due to the foreground being located in different sections of the images, the average image would often resemble a uniform baseline.

## 2.3 ONE-VS-ONE AND ONE-VS-ALL ATTRIBUTION

In multi-class settings, while one-vs-all explanation $A_{S,c_o}(x)$ was designed to explain why the input $x$ belong to its original class $c_o$ and not the others, one-vs-one explanations aim to provide an attribution $A_{S,c_o \to c_t}(x) \in \mathbb{R}^N$ that explains why $x$ belong to $c_o$ and not the specified target class $c_t$. Most existing attribution methods were primarily designed for one-vs-all explanation, but was proposed to extend to one-vs-one by simply calculating the attribution with respect to the difference of the original class probability to the target class probability $S_{\text{diff}}(x) = S_{c_o}(x) - S_{c_t}(x)$ (Bach et al. (2015); Shrikumar et al. (2017)).

It is easy to think of examples where this somewhat naive formulation will not provide correct one-vs-one explanation. Taking the example of fruit classification, for both apples and oranges the explanation could easily be the round shape, and taking the difference between those will result in an arbitrary attribution. We claim that without a class-targeted baseline, the modified attributions will still omit the "vs-one" aspect of the one-vs-one explanation. Take IG for example, $A_{S,\text{diff}}(x) = A_{S,c_o}(x) - A_{S,c_t}(x)$. With zero baseline, the target class score $S_{c_t}(x)$ and its gradient will likely stay close to zero along the straight path from the input to the zero baseline, meaning that $A_{S,c_t}(x) \approx 0$ because the instance never belongs to the target class. With this in mind the one-vs-one explanation is not very informative with respect to the target class $c_t$.

Few class-targeted baselines were proposed in the past. The minimum distance training sample (MDTS) described in Section 2.2 is class-targeted as the sample was selected from the designated target class. While the original expected gradient was defined for one-vs-all explanation only, we extended the method to one-vs-one by sampling the baselines only from the target class. However, as mentioned in Section 2.2, MDTS is frequently hindered by the sparsity of the training set in the high dimensional space, and expected gradient suffers from undesired effects caused by uncorrelated training samples. The problem of baseline selection, especially for the one-vs-one explainability, has

presented a challenging problem, because the ideal baseline choice can simply be absent from the training set.

## 2.4 GAN AND IMAGE-TO-IMAGE TRANSLATION

Image-to-Image Translation is a family of GAN originally introduced by Isola et al. (2017) for creating mappings between two domains of data. While the corresponding pairs of images are rare in most real-world dataset, Zhu et al. (2017) has made the idea widely applicable by introducing a reconstruction loss to tackle the tasks with unpaired training dataset. Since then, more efficient and better performing approaches have been developed to improve few-shot performance (Liu et al. (2019)) and output diversity (Choi et al. (2020)). Nevertheless, we found the StarGAN variant proposed by Choi et al. (2017) specifically applicable to the baseline selection problem because of its standalone class discriminator in the adversarial networks as well as the deterministic mapping that preserve the styles of the translated images (Choi et al. (2020)). GANs have not been applied for explaining DNNs in the past to our best knowledge.

Prior to our work, Chang et al. (2018) proposed the fill-in the dropout region (FIDO) methods and suggested generators including CA-GAN Yu et al. (2018) for filling in the masked area. However, the CA-GAN generation was designed for calculating the smallest sufficient region and smallest destroying region Dabkowski & Gal (2017) that only produced 1-vs-all explanations. FIDO is computationally expensive as an optimization task is required for each attribution map. The fill-in method requires an unmasked area for reference, hence only works for a small subset of attribution methods. More importantly, the FIDO is highly dependent on the generator's capability of recreating the image based on partially masked features. With pre-trained generators like CA-GAN, we argue that the resulting saliency map is more associated with the pre-trained generator instead of the classifier itself.

## 3 GAN-BASED MODEL EXPLAINABILITY

It has been previously established that attribution methods are more sensitive to features where the input values are the same as the baseline values and less sensitive to those where the input values and the baseline values are different (Adebayo et al. (2018); Sundararajan & Taly (2018)). Therefore, we expect a well-chosen baseline to differ from the input only on the key features. Good candidates for achieving this would be the sample in the target class but with minimum distance to the input.

Formally, for a one-vs-one attribution problem $A_{S,c_o \to c_t}(x)$, We define the class-targeted baseline to be the closest point in the input space (not limited to the train set) that belongs to the target class

$$B_{c_t}(x) = \arg \min_{\tilde{x} \in G_{c_t}} \|x - \tilde{x}\| \tag{1}$$

Here, $G_{c_t}$ is the set of realistic examples in the target class, and $\| \cdot \|$ is the Euclidean distance. By using this baseline we have $A_{S,c_o \to c_t}(x, B_{c_t}(x))$ providing the explanations as to why input $x$ belongs to its original class $c_o$ and not class $c_t$. Now, since it isn't realistic to optimize within the actual set $G_{c_t}$ we work with a softer version of Equation 1: $B_{c_t}(x) = \arg \min_{\tilde{x} \in \mathbb{R}^N} (\|x - \tilde{x}\| - \log T(\tilde{x}, c_t))$. where $T(\tilde{x}, c_t)$ represent the probability of $\tilde{x}$ belonging to the target class, meaning $\tilde{x} \in G_{c_t}$. Given a classifier $S(x) = [S_1(x), ..., S_C(x)]$, we have the estimate $S_c(\tilde{x})$ to the probability of a realistic image $\tilde{x}$ to be in class $c$. In order to make use of this we decompose $T(\tilde{x}, c_t) = R(\tilde{x})S_{c_t}(\tilde{x})$ where $R(\tilde{x})$ indicates the probability of $\tilde{x}$ being a realistic image. We end up with the following objective for the baseline instance.

$$B_{c_t}(x) = \arg \min_{\tilde{x} \in \mathbb{R}^N} (\|x - \tilde{x}\| - \log R(\tilde{x}) - \log S_{c_t}(\tilde{x})) \tag{2}$$

### 3.1 APPLYING STARGAN TO THE CLASS-TARGETED BASELINE

Here we introduce GAN-based Model EXplainability (GANMEX) that uses GAN to generate the class-targeted baselines. Given an input $x$ and a target class $c_t$, GANMEX aims to generate a class-targeted baseline $G(x, c_t)$ that achieve the three following objectives:

1. The baseline belongs to the target class (with respect to the classifier).
2. The baseline is a realistic sample.

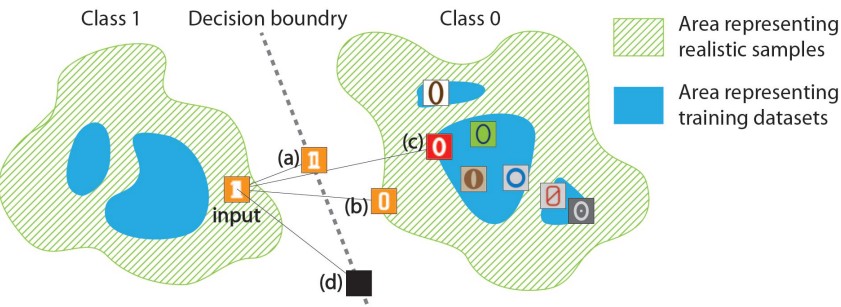

Figure 1: Intuition of using GANs for generating class-targeted baselines in SVHN dataset. Without GANs, a closest target class sample can easily be unrealistic (a), while the GAN helps confine the sample in the realistic sample space (b). The MDTS baseline and other training samples used in expected gradient can be very different from the input (c). (d) shows the zero baseline that is the most commonly used.

 3. The baseline is close to the input.

To further explain the need for all 3 objectives, we point the reader to Figure 1. The GANMEX baseline represents the "closest and realistic target class baseline". Without the assistance of GANs, the selected baseline can easily either fall into the domain of unrealistic image. A naive fix will choose a realistic image from the training set, but that will not be close to the input. Finally, for correct one-vs-one explainability we need the baseline to belong to a specific target class. We have provided more intuitions behind the baseline selection requirements in Appendix G.

We chose StarGAN (Choi et al. (2017)) as the method for computing the above $T$ or rather $R$ function. Although many Image-to-Image translation methods could be applied to do so, StarGAN inherently works with multi-class problems, and allows for a natural way of using the already trained classifier $S$ as a discriminator, rather than having us train a different discriminator.

StarGAN provides a scalable image-to-image translation approach by introducing (1) a single generator $G(x, c)$ accepting an instance $x$ and a class $c$, and producing a realistic example $x$ in the target class $c$, (2) two separate discriminators: $D_{\text{src}}(x)$ for distinguishing between real and fake images, and $D_{\text{cls}}(x, c)$ for distinguishing whether $x$ belongs to class $c$. It introduced following loss functions

$$
\begin{aligned}
\mathcal{L}_{\text{adv}} &= \mathrm{E}_x[\log(D_{\text{src}}(x))] + \mathrm{E}_{x,c}[\log(1 - D_{\text{src}}(G(x, c)))] & (3) \\
\mathcal{L}_{\text{cls}}^r &= \mathrm{E}_{c',x \in c'}[-\log(D_{\text{cls}}(c'|x))] & (4) \\
\mathcal{L}_{\text{cls}}^f &= \mathrm{E}_{x,c}[-\log(D_{\text{cls}}(c|G(x, c)))] & (5) \\
\mathcal{L}_{\text{rec}} &= \mathrm{E}_{c,c',x \in c'}[\|x - G(G(x, c), c')\|_1] & (6)
\end{aligned}
$$

Here, $\mathrm{E}.[]$ defines the average over the variables in the subscript, where $x$ is an example in the training set, and $c, c'$ are classes. $\mathcal{L}_{\text{adv}}$ is the standard adversarial loss function between the generator and the discriminators, $\mathcal{L}_{\text{cls}}^r$ and $\mathcal{L}_{\text{cls}}^f$ are domain classification loss functions for real images and fake images, respectively, and $\mathcal{L}_{\text{rec}}$ is the reconstruction loss commonly used for unpaired image-to-image translation to make sure two opposite generation action will lead to the original input. The combined loss functions for the generator and the discriminator are

$$
\begin{aligned}
\mathcal{L}_D &= -\mathcal{L}_{\text{adv}} + \lambda_{\text{cls}}^r \mathcal{L}_{\text{cls}}^r & (7) \\
\mathcal{L}_G &= \mathcal{L}_{\text{adv}} + \lambda_{\text{cls}}^f \mathcal{L}_{\text{cls}}^f + \lambda_{\text{rec}} \mathcal{L}_{\text{rec}} & (8)
\end{aligned}
$$

The optimization procedure for StarGAN alternates between modifying the discriminators $D_{\text{src}}(x)$, $D_{\text{cls}}(x, c)$ to minimize $\mathcal{L}_D$, and the generator $G$ to minimize $\mathcal{L}_G$.

Equation 8 is almost analogical to equation 2. The term $\mathcal{L}_{\text{adv}}$ corresponds to $-\log R(\tilde{x})$ and the term $\lambda_{\text{cls}}^f \mathcal{L}_{\text{cls}}^f$ corresponds to the term $\log S_{c_t}(\tilde{x})$. There is a mismatch between the term $\lambda_{\text{rec}} \mathcal{L}_{\text{rec}}$ and $\|x - \tilde{x}\|$. One forces the generator to be invertible, while the other forces the generated image to be close to the original. We found that the $\mathcal{L}_{\text{rec}}$ term is useful to encourage the convergence of the GAN. However, a similarity term $\|x - G(x, c)\|$ is also needed in order for the baseline image to be close to the origin - this allows for better explainability. We show in what follows (Figure 7.B,

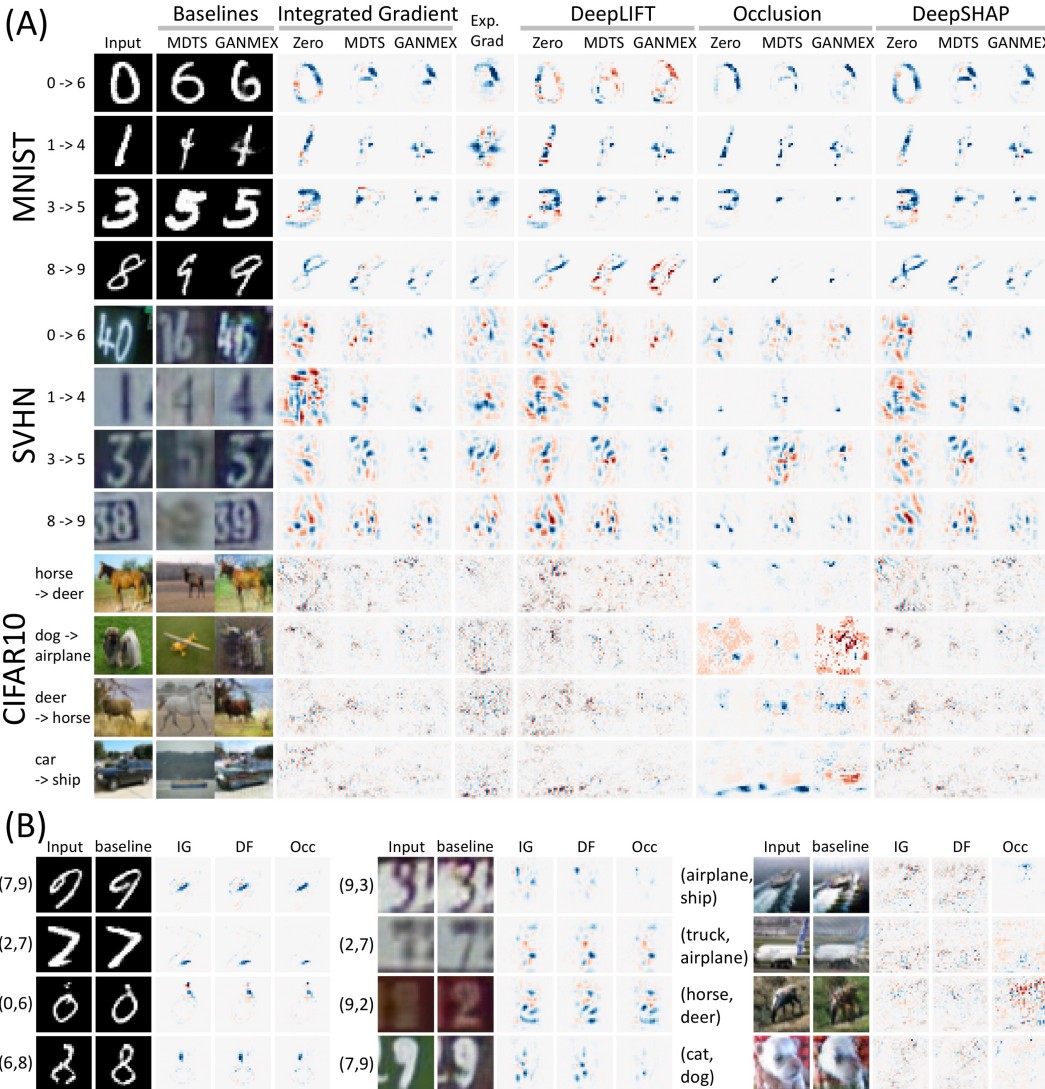

Figure 2: (A) Saliency maps for multi-class datasets (MNIST, SVHN, CIFAR10) generated with various baselines, including zero baseline (Zero), MDTS and GANMEX, with classes $c_o \rightarrow c_t$ indicated for each example. (B) Mis-classification analysis showing with the (mis-classified class, correct class) pairs. The baseline columns show the expected images generated by GANMEX for the correct classes, and the saliency maps show the explanation of "why not the correct class" produce by IG, DeepLIFT (DL) and occlusion (Occ).

Appendix D) that without this similarty term, the created image can indeed be farther away from the origin. Other than the added similarity term, for GANMEX we replace the discriminator $D_{\mathrm{cls}}(c|\tilde{x})$ with the classifier $S_c(\tilde{x})$, since as mentioned above, this way the generator provides a baseline adapted to our classifier. Concluding, we optimize the following term for the generator

$$\mathcal{L}_G^f = \log(1 - D_{\mathrm{src}}(\tilde{x})) - \lambda_{\mathrm{cls}}^f \log(S_c(\tilde{x})) + \lambda_{\mathrm{rec}}\|x - G(\tilde{x}, c')\|_1 + \lambda_{\mathrm{sim}}\|x - \tilde{x}\|_1 \qquad (9)$$

where $\tilde{x}$ is short for $G(x, c)$. Notice that we used L1 distance rather than L2 for the similarity loss, because L2 distance leads to blurring outputs for image-to-image translation algorithms (Isola et al. (2017)). Other image-to-image translation approaches can potentially select baselines satisfying the criteria (2) and (3) above, but they lack the replaceable class discriminator component, that is crucial for explaining the already trained classifier. We provide several ablation studies in Appendix D where we show that without incorporating the to-be-explained classifier to the adversarial networks, the GAN generated baselines will fail the randomization sanity checks. We provide more implementation details including hyper-parameters in Appendix A.2.

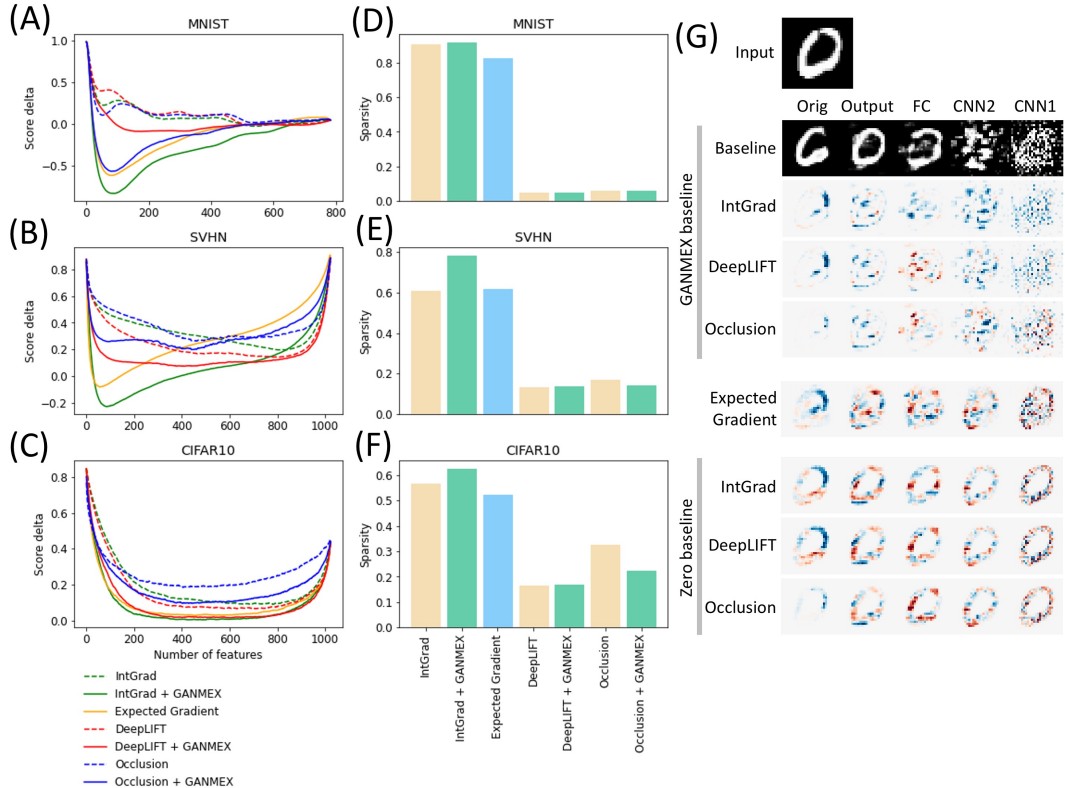

Figure 3: (A-C) Perturbation-based evaluation plots for MNIST and SVHN, respectively. The dashed lines represent the non-class-targeted baselines and the solid lines represent class-targeted baselines. (D-F) Gini indices, with the yellow bars represent saliency maps with zero baselines and the green bars represent that of GANMEX baselines. (G) Sanity checks showing the original saliency maps (Orig) and saliency maps under cascading randomization over the four layers: output layer (Output), fully connected (FC), and two CNN layers (CNN1, CNN2).

## 4  EXPERIMENTS

In what follows we experiment with the datasets **MNIST** (LeCun & Cortes (2010)), **Street-View House Numbers (SVHN)** (Netzer et al. (2011)), **CIFAR10** (Krizhevsky (2009)) and **apple2orange** (Zhu et al. (2017)). Further details about the datasets and classifiers are given in Appendix A.1.

Our techniques are designed to improve any global attribution method by providing an improved baseline. In our experiments we consider four attribution methods - IG, DeepLIFT, Occlusion, and DeepSHAP. The baselines we consider include the zero baseline (the default baseline in all four methods), minimum distance training sample (MDTS), and the GANMEX baseline. We also compared our results with a modified version of expected gradient aimed to provide 1-vs-1 explanations, which runs IG over a randomly chosen target class image from the training set, as opposed to a random image from the training set.

### 4.1  ONE-VS-ONE ATTRIBUTION FOR MULTI-CLASS CLASSIFIERS

We tested the one-vs-one attribution on two multi-class datasets - MNIST, SVHN, and CIFAR10. As shown in Figure 2.A, the GANMEX baseline successfully identified the closest transformed image in the target class as the baseline. Take explaining why 0 and not 6 for example, the ideal baseline would keep the "C"-shape part unchanged, and only erase the top-right corner and complete the lower circle, which was achieved by GANMEX. Limited by the training space, MDTS baselines were generally more different from the input image. Therefore, the explanation made with respect to GANMEX baselines were more focused on the key features compared to that of the MDTS baseline and expected gradient. We observed the same trends across more numbers, where GANMEX helps IG, DeepLIFT, Occlusion and DeepSHAP disregard the common strokes between the original

and targeted digits, and focusing only on the key differences. The out-performance of GANMEX was even more obvious in the SVHN datasets, where the numbers can have any font, color, and background, and in CIFAR10, which has more complexity and diversity. Notice that both the zero baseline and training set baseline cause the explanation to have more focus on the background, and in contrast, the GANMEX example focuses only on the key features that would cause the digit to change.

Zero baselines, on the other hand, were generally unsuccessful in making one-vs-one explanations. The attributions on MNIST look similar to the original input and ignores everything in the background, and the attributions on SVHN were rather noisy. As shown in Figure 8, attributions based on zero baselines only changed marginally with different target classes. This shows that purposely designed class-targeted baselines are required for meaningful one-vs-one explanation.

**Mis-classification Analysis**   We next demonstrated how one-vs-one saliency maps can be used for trouble-shooting mis-classification cases. For an input $x$ that belongs to class $c_o$ but was mis-classified as class $c_m$. $A_{S,c_m \rightarrow c_o}(x) = A_{S,c_m \rightarrow c_o}(x, B_{c_o}(x))$ provides explanation to why $x$ belongs to $c_m$ and not $c_o$ with respect to the trained classifier, and this will help human understand how the classifier has led to the incorrect decision. We provided examples in Figure 2.B where the samples were mis-classified. For MNIST and SVHN samples, the mis-classification mostly happened when the digits were presented in a non-typical way. The GANMEX baseline $B_{c_o}(x)$ show how a more typical digit should have been written according to the trained classifier, and the attribution $A_{S,c_m \rightarrow c_o}(x)$ highlights the area that led to the mis-classification. CIFAR10 presented more complex classification challenges, and the classifier can easier confused ship with an airplane because of the pointy front and the lack of sea horizon, or a dog with a cat because of the shape of the ears and the nose, and those areas were highlighted in the one-vs-one saliency maps.

**Perturbation-based evaluation**   We followed the perturbation-based evaluation suggested by Bach et al. (2015) that flips input features starting from the ones with the highest saliency values and evaluates the cumulative impacts on the score delta $S_{c_o} - S_{c_t}$ as proposed by Shrikumar et al. (2017). Flipping a feature means to provide with a value of $1 - x$ where $x$ is its original value, assuming all features are normalized to $x \in [0, 1]$. A wanted behavior from the attribution map is that the score delta will decrease as rapidly as possible as we flip the features one by one. We provide in Figure 3.A-C the perturbation curves for both MNIST, SVHN and CIFAR10, plotting the score delta as a function of the number of flipped features. It is painfully clear that that by using a GANMEX baseline rather than the alternative zero baseline, the descent of the curve is much faster, meaning that we successfully capture the most important features using GANMEX. This holds true for all attribution methods. As a side note, notice that in SVHN when we flip all features the score delta goes back to where it was in the beginning as opposed to going down to zero. This is due to the fact that once all features are flipped, we are back to having the same digit as before.

**Gini-index**   We calculated the Gini Index representing the sparseness of the saliency maps as proposed by Chalasani et al. (2018), where a larger score means sparser saliency map, which is a desired property. Figure 3.D-F shows our experiments comparing the different techniques with a zero baseline vs. GANMEX. Other than IG/SVHN where GANMEX has a visible advantage, the results are roughly the same; we suspect that the sparseness of zero baseline attribution was benefited from incorrectly hiding key features, as shown in Figure 2.A and 3.A-C. Expected gradient, on the other hand, consistently under-performs its counterpart (IG+GANMEX) in both datasets.

**Cascading randomization**   We performed the sanity checks proposed by Adebayo et al. (2018) that performs cascading randomization from top to bottom layers of the DNN and observe the changes in the saliency maps. Specifically, layer by layer we replace the model weights with Gaussians random variables scaled to have the same norm. For meaningful model explanations, we would expect the attributions to be gradually randomized during the cascading randomization process. In contrast, unperturbed saliency maps during the model randomization would suggest that the attributions were based on general features of the input and not specifically based on the trained model.

Figure 3.G shows the experiment on MNIST data with the network layers named (input to output) CNN1, CNN2, FC, Output. It shows that even though the saliency maps generated by the original IG, DeepLIFT and Occlusion were rather unperturbed (still showing the shape of the digit) after the

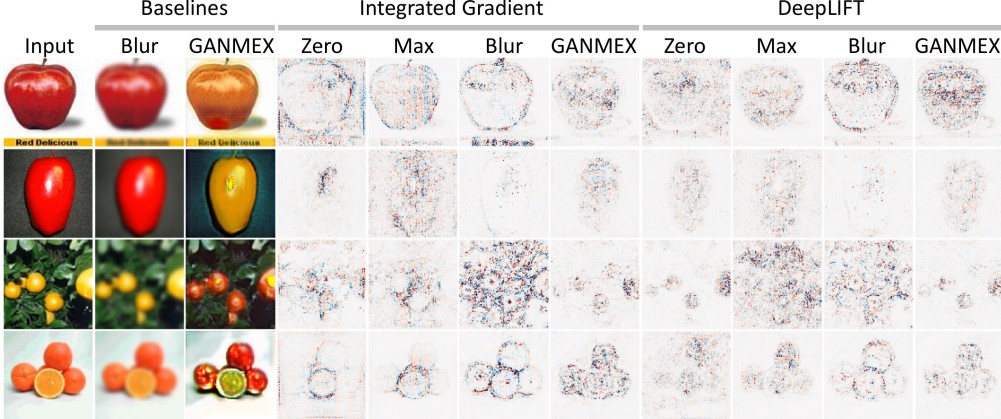

Figure 4: Saliency maps for the classifier on the apple2orange dataset with four baseline choices: zero baseline (Zero), maximum value baseline (Max), blurred baseline (Blur), and GANMEX baseline (GANMEX).

model randomization, with the help of GANMEX, both the baselines and the saliency maps were perturbed over the cascading randomization. Expected gradient, while showing more randomization compared to the zero baseline saliency maps, still roughly shows the shape of the digit throughout the sanity check.

## 4.2 ATTRIBUTION FOR BINARY CLASSIFIERS

In addition to the one-vs-one aspect, GANMEX generally improves the quality of the saliency maps compared with the existing baselines, and this can be tested on binary datasets where the one-vs-one explanations and the one-vs-all explanations are equivalent. For apple2orange dataset, conceptually apples and oranges both have round shapes but have different colors, so we would expect the saliency maps on a reasonably performing classifier to highlight the colors of the fruit, but not the shapes, and definitely not the background.

In Figure 4 and Figure 9 we compared the saliency map generated by DeepLIFT and IG with the zero input, max input, blurred image, with those generated with the GANMEX baselines. In all non-GANMEX baselines (Zero, Max, Blur) we commonly observe one of two errors in the saliency map. The first error consists of highlighting the background. The second highlights only the edge of the apple(s) providing the false indication that the model is basing its decisions on the shape of the object rather than its color. It is quite clear that neither of these errors occur when using the GANMEX baselines as the background is never present and the full shape of the apple(s)/orange(s) is highlighted.

## 5 CONCLUSION AND FUTURE WORK

We have proposed GAN-based model explainability, a novel approach for generating one-vs-one explanation baselines without being constrained by the training set. We used the GANMEX baselines in conjunction with IG, DeepLIFT, SHAP, and Occlusion, and to our surprise, the baseline replacement was all it takes to address the common downside of the existing attribution methods (blind to certain input values and fail to randomize with the model randomization) and significantly improve the one-vs-one explainability. The out-performance was demonstrated through perturbation-based evaluation, sparseness measures, and cascading randomization sanity checks. The one-vs-one explanation achieved by GANMEX opens up possibilities for obtaining more insights about how DNNs differentiate similar classes.

While GANMEX showed promising results on explaining binary classifiers, where one-vs-all and one-vs-one explanations are directly comparable, open questions remain on how to apply GANMEX to one-vs-all explainability for multi-class classifiers, and how to best optimize the GAN component to effective generate baselines for classification tasks with large number of classes.

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

# A IMPLEMENTATION DETAILS

## A.1 DATASETS AND CLASSIFIERS

**MNIST** (LeCun & Cortes (2010)) The classifier consists of two 6x6 CNN layers with a stride of 2, followed by a 256-unit fully connected layer, a dropout layer with $p = 0.5$, and the 10 output neurons. As shown in Springenberg et al. (2015) the stride>1 CNN achieved comparable performance with pooling layers. The classifier was trained for 50 epochs and achieve a test accuracy of 99.3%.

**Street-View House Numbers (SVHN)** (Netzer et al. (2011)) We tested our models on the cropped version of SVHN and used the same model architecture with that of MNIST and achieved a test accuracy of 90.3% after 50 epochs of training.

**CIFAR10** (Krizhevsky (2009)) We trained a classifier consist of 4 repetitive units, with each unit constructed by two 3x3 CNN layers and a 2x2 average pooling layer, with each CNN layer followed by a batch normalization layer. The classifer achieved 87.8% test accuracy after 100 epochs of training.

**apple2orange** (Zhu et al. (2017)) We trained a classifier taking the original 256x256 image as input. The classifier was constructed by adding a global average pooling layer on top of MobileNet (Howard et al. (2017)), and then followed by a dense layer of 1024 neurons and a dropout layer of $p = 0.5$ before the output neurons. The classifier was trained for 50 epochs and achieve a test accuracy of 87.7%.

## A.2 BASELINE GENERATION WITH GANMEX

Our baseline generation process is based on StarGAN (Choi et al. (2017)). We used the Tensorflow-GAN implementation (https://github.com/tensorflow/gan) and made the following two modifications (Equation 9):

1. The class discriminator $D_{\text{cls}}$ is replace by the target classifier $S$ to be explained.
2. A similarity loss $\mathcal{L}_{sim}$ is added to the training objective function.

We train the GANMEX model for 100k steps for the MNIST and apple2orange datasets, 300k steps for the SVHN dataset, and 400k steps for the CIFAR10 dataset. Only the train split is used for training, and the attribution results and evaluation were done on the test split of the dataset.

## A.3 ATTRIBUTION METHODS

We used DeepExplain (https://github.com/marcoancona/DeepExplain) for generating saliency maps with IG, DeepLIFT, and Occlusion. We modified the code base to use the score delta ($S_{c_o} - S_{c_t}$) instead of the original class score ($S_{c_t}$) and allowing replacing the zero baseline (see Section 2.1) by custom baselines from GANMEX and MDTS. Expected gradient was separately implemented according to the formulation in Erion et al. (2019). We set the number of sampling steps to 200 for both IG and Expected Gradient, and used Occlusion-1 that only perturb the pixel itself (as supposed to perturbing the whole neighboring patch of pixels).

The DeepSHAP saliency maps were calculated using SHAP (https://github.com/slundberg/shap). We made similar modification to replace the original class score by the score delta and feed in the custom baseline instances.

In all saliency maps shown in the paper, blue color in indicates positive values and red color indicates negative values. We skipped Occlusion for large images (apple2orange) and also skipped SHAP for full dataset evaluations due to the computation resource constraints.

# B BASELINE DISTANCE ANALYSIS

To measure how various baseline selection approaches satisfy the minimum distance requirements in Equation 1, we calculated $D(x, \tilde{x}) = \|x - \tilde{x}\|$ for (1) GANMEX, (2) MDTS, (3) a randomly selected sample in the target class as baseline and (4) zero baseline. GANMEX was on-par with MDTS on the MNIST dataset, but on SVHN and CIFAR10 dataset that have more degrees of freedom (object

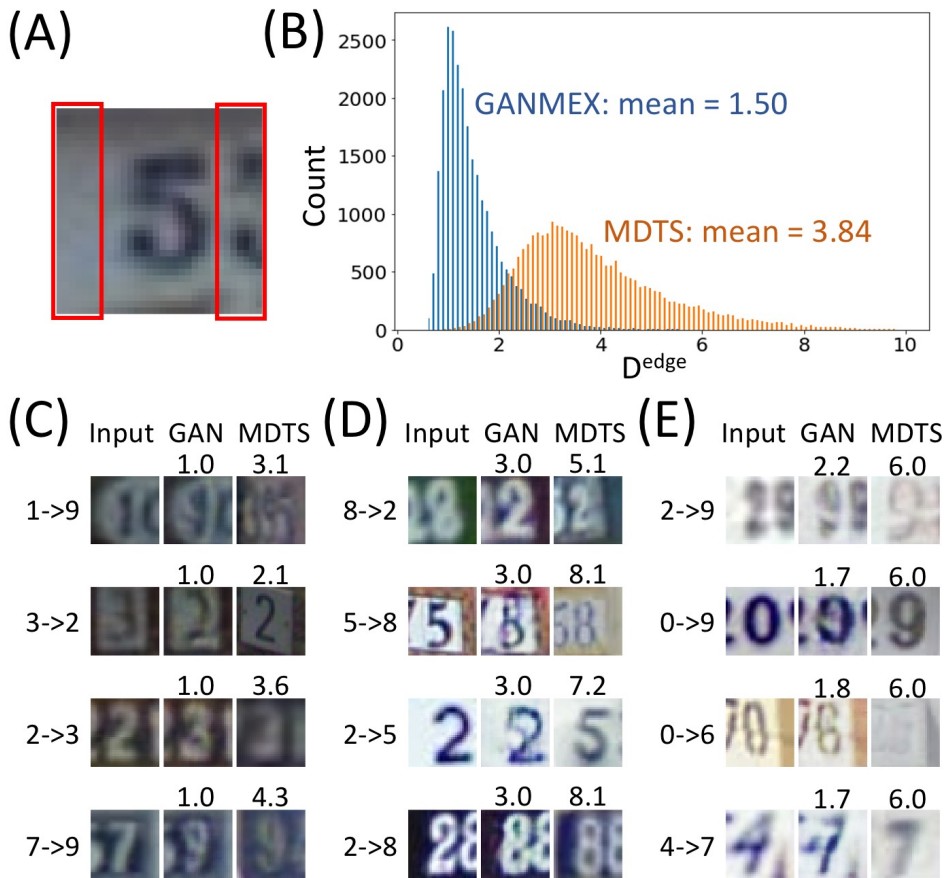

Figure 5: (A) Vertical edge area in an SVHN image. (B) Histogram of sample to baseline distance ($D^{\mathrm{edge}}(x, \tilde{x})$) in the vertical edge area. (C-E) Samples comparing GANMEX and MDTS baselines with $D^{\mathrm{edge}}(x, \tilde{x})$ indicated on the top of the baseline images. (C) Easy cases for GANMEX ($D^{\mathrm{edge}}(x, \tilde{x}) \approx 1$). (D) Difficult cases for GANMEX ($D^{\mathrm{edge}}(x, \tilde{x}) \approx 3$). (E) Difficult cases for GANMEX ($D^{\mathrm{edge}}(x, \tilde{x}) \approx 6$).

Table 1: Baseline distance analysis comparing the average intra-class distance ($D_{\mathrm{intra}}$, sampled) and the average inter-class distance ($D_{\mathrm{inter}}$, sampled), with the average distance from the instance to the baseline input generated by GANMEX (GAN), MDTS, random selection (RAND), and zero inputs (Zero).

| | | Data Size | | Avg. Distance | | Baseline Distance | | | |
|---|---|---|---|---|---|---|---|---|---|
| | Dimension | Train | Test | $D_{\mathrm{intra}}$ | $D_{\mathrm{inter}}$ | GAN | MDTS | Rand | Zero |
| MNIST | 784 (28 x 28 x 1) | 60,000 | 10,000 | 8.96 | 10.32 | **7.17** | 7.18 | 10.32 | 9.28 |
| SVHN | 3072 (32 x 32 x 3) | 73,257 | 26,032 | 14.42 | 14.48 | **3.42** | 5.94 | 15.44 | 26.52 |
| CIFAR10 | 3072 (32 x 32 x 3) | 50,000 | 10,000 | 18.28 | 19.06 | **6.89** | 10.54 | 19.01 | 29.04 |

size, color, orientation, background, ...), GANMEX was significantly better in identifying minimum distance baselines compared to the in-sample search. The high dataset complexity of was supported by the average intra-class distance, the average distance between any two instances within the same class, which was higher than that of MNIST. Note that the resulting sample to baseline distance $D(x, \tilde{x}_{\text{GANMEX}})$ is much higher in MNIST than in SVHN, because there were more boundary values (0s and 1s) in MNIST.

We further evaluated the similarity distance on the vertical edge area ($D^{\text{edge}}(x, \tilde{x})$) of the SVHN images (Figure5.A). Empirically, we observed that the digits of interest were rarely present in the vertical edge, and therefore, we would expect a closest baseline choice will lead to minimal $D^{\text{edge}}(x, \tilde{x})$ under the minimum distance requirements. We provided a histogram in Figure 5.B for comparing the distribution of $D^{\text{edge}}(x, \tilde{x})$ for MDTS and GANMEX, and we presented sampled success/failure cases in Figure 5.C-E. Overall, GANMEX leads to baselines that are closer to the original samples.

## C  ADDITIONAL METRICS

Here we evaluated the different baseline choices with additional metrics (Table 2). $\text{AOPC}_L$ measures the area over the perturbation curve within the first $L$ perturbation steps (Samek et al. (2017); Tomsett et al. (2019)). One potential downside of $\text{AOPC}_L$ is that the metric is only sensitive to the top $L$ features in the saliency map and not the rest. Therefore, in addition to $\text{AOPC}_{L=100}$, we calculated $\text{AOPC}_{\text{all}}$, the area over the perturbation curve across all feature. The gradient family of IG and expected gradient generally outperformed DeepLIFT and occlusion on the AOPC metrics, with IG+GANMEX performed the best overall.

The sparsity (Chalasani et al. (2018)) measured by the Gini index is a desirable property for one-vs-one attribution. We expect a good one-vs-one explanation to highlight only the differentiating features. Compared with one-vs-all saliency maps, one-vs-one saliency maps should highlight a smaller subset of features, especially when the target classes are similar to the original classes. Therefore, one would expect more sparse one-vs-one saliency maps are more likely to be correct. Our results showed that saliency maps generated by IG+GANMEX and IG+MDTS have higher Gini index and therefore are more sparse compared to other methods (Table 2).

We also measured the faithfulness reported by Alvarez-Melis & Jaakkola (2018); Tomsett et al. (2019) and monotonicity suggested by phi Nguyen & Martínez (2020). Instead of measuring the cumulative effect of alternating a set of features, both faithfulness and monotonicity measure the impacts on alternating single features. We found that expected gradient and Occlusion+zero baseline are the best performers on those two metrics.

Lastly, we designed an inverse localization metric to measure whether the explanation is localized in the focus area. We observed that the digits mostly have $< 1$ aspect ratios, meaning that their widths are smaller then their heights. As a results, the areas at the two vertical edges are generally not covered by the primary numbers, and instead, they usually show the background or the neighboring numbers. Therefore, we can reasonably expect the saliency map sensitivity to be location in the center area (area excluding the vertical edges), and not the vertical edge area (Figure 5.A).

Based on this observation, we designed the inverse localization metric, $L(A(x)) = (\frac{1}{card(S_{\text{edge}})} \sum_{i \in S_{\text{edge}}} |A_i(x)|)/(\frac{1}{card(S_{\text{center}})} \sum_{i \in S_{\text{center}}} |A_i(x)|)$, that calculates the ratio of the average absolute sensitivity between the vertical edge area and the center area. $S_{\text{center}}$ and $S_{\text{edge}}$ represent the feature set in the center area and the edge area, respectively. $card(.)$ measures the cardinality of feature sets, and $x$ and $A(x)$ are the sample and the corresponding saliency map. A lower $L(A(x))$ would mean that the saliency map $A(x)$ is more localized in the focus area.

As shown in Table 2, we see a consistent trend of the saliency maps with GANMEX baselines being more localized (lower inverse localization) compared to MDTS baselines across all attribution methods, and expected gradient and the zero baselines generally lead to the worst results. The saliency maps produced by occlusion+GANMEX was the most localized among all the methods tested.

To summarize, we evaluated multiple attribution methods and baseline combinations with metrics that assess different properties of the saliency maps. While the results of faithfulness and monotinicity were less consistent, GANMEX has consistently lead to better metrics on inverse localization,

Table 2: Additional metrics for attribution methods using the zero baseline (Zero), MDTS, and GANMEX (GAN).

| Metrics | Dataset | Integrated Gradient | | | EG | DeepLIFT | | | Occlusion | | |
|---|---|---|---|---|---|---|---|---|---|---|---|
| | | Zero | MDTS | GAN | | Zero | MDTS | GAN | Zero | MDTS | GAN |
| AOPC$_{100}$ | MIST | 0.614 | 1.249 | **1.421** | 1.260 | 0.505 | 0.639 | 0.724 | 0.705 | 1.050 | 1.221 |
| | SVHN | 0.346 | 0.861 | **0.921** | 0.878 | 0.377 | 0.634 | 0.621 | 0.317 | 0.547 | 0.549 |
| | CIFAR10 | 0.298 | 0.485 | 0.494 | **0.516** | 0.323 | 0.464 | 0.451 | 0.441 | 0.492 | 0.440 |
| AOPC$_{all}$ | MIST | 0.889 | 1.098 | **1.263** | 1.13 | 0.859 | 0.933 | 0.992 | 0.877 | 1.019 | 1.114 |
| | SVHN | 0.564 | **0.844** | 0.822 | 0.626 | 0.649 | 0.750 | 0.751 | 0.528 | 0.586 | 0.585 |
| | CIFAR10 | 0.696 | 0.788 | **0.808** | 0.789 | 0.726 | 0.780 | 0.794 | 0.628 | 0.694 | 0.706 |
| sparsity | MIST | 0.909 | 0.911 | **0.919** | 0.827 | 0.047 | 0.047 | 0.046 | 0.062 | 0.058 | 0.058 |
| | SVHN | 0.606 | 0.713 | **0.783** | 0.615 | 0.131 | 0.133 | 0.139 | 0.168 | 0.139 | 0.144 |
| | CIFAR10 | 0.565 | **0.639** | 0.626 | 0.522 | 0.164 | 0.171 | 0.169 | 0.325 | 0.260 | 0.223 |
| faithfulness | MIST | 0.182 | 0.224 | 0.280 | **0.407** | 0.075 | 0.003 | 0.031 | 0.257 | 0.254 | 0.291 |
| | SVHN | 0.017 | 0.265 | 0.270 | **0.548** | -0.041 | 0.075 | 0.017 | 0.007 | 0.306 | 0.243 |
| | CIFAR10 | 0.005 | 0.028 | 0.027 | 0.054 | 0.003 | 0.017 | 0.017 | **0.288** | 0.285 | 0.225 |
| monotonicity | MIST | 0.118 | 0.196 | 0.264 | **0.357** | 0.087 | 0.150 | 0.206 | 0.244 | 0.239 | 0.280 |
| | SVHN | 0.129 | 0.212 | 0.248 | **0.340** | 0.095 | 0.150 | 0.182 | 0.057 | 0.210 | 0.211 |
| | CIFAR10 | 0.008 | 0.050 | 0.042 | 0.058 | 0.004 | 0.044 | 0.033 | **0.175** | 0.140 | 0.087 |
| inv. localization | SVHN | 0.268 | 0.128 | 0.113 | 0.217 | 0.268 | 0.156 | 0.123 | 0.268 | 0.144 | **0.100** |

AOPC$_L$, AOPC$_{all}$ and sparsity, especially when compared with the non-class-targeted zero baseline that were widely used in the field.

# D   ABLATION STUDIES

Here we analyzed the possibilities of using other GAN models. Different from StarGAN, most other image-to-image translation algorithms do not have a stand-alone class discriminator that can be swapped with a trained classifier. To simulate such restrictions, we trained a similar GAN model but with the class discriminator trained jointly with the generator from scratch. Figure 7.A shows that while the stand-alone GAN yields similar baseline with GANMEX, both the baselines and saliency maps of the stand-alone GAN remains unperturbed under cascading randomization of the model. This indicates that the class-wise explanations provided by stand-alone GAN were not specific to the to-be-explained classifier.

The importance of the similarity loss in Equation 9 can be demonstrated on a colored-MNIST dataset, where we randomly assigned the digits with one of the three colors {red, green, blue}, with labels of the instances remain unchanged from the original MNIST labels of $\{0, ..., 9\}$. The classifier was trained with the same model architecture and training process as for MNIST.

The dataset demonstrated different modes (colors in this case) that are irrelevant to the labels, and we would expect the class-targeted baseline for $x$ would be another instance that has the same color as $x$. Figure 7.B shows that the similarity loss is the crucial component for ensuring that the baseline has the same color with the input. Without the similarity loss, the generated baseline instance can easily have a different color with the original image. The reconstruction loss itself does not provide the same-mode constraint because a mapping of $G(\text{red}) \rightarrow$ green and $G(\text{green}) \rightarrow$ red does not get penalized by the reconstruction loss. While the reconstruction loss was not required for GANMEX and the same-mode constraint, we observed that some degrees of reconstruction loss help GANs converge faster.

# E   HYPER-PARAMETER ANALYSIS

We tested how the generated baselines change with respect to the hyperparemeters in the GANMEX loss function. The hyper-parameters, $\lambda_{cls}^{f}$, $\lambda_{rec}$, and $\lambda_{sim}$, presented in Equation 9 control the degrees of the classification loss, reconstruction loss, and similarity loss, respectively. We performed the hyper-parameter scan on the SVHN dataset as it has enough complex and yet simple enough for visually assessing the attribution.

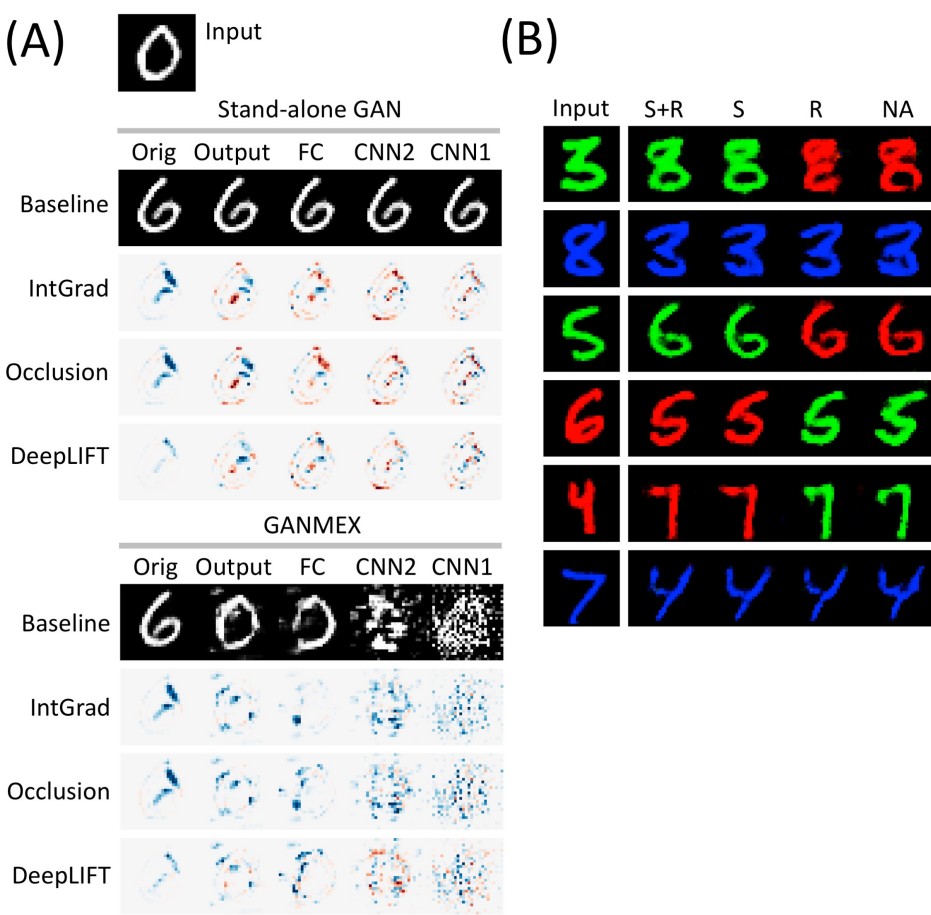

Figure 6: (A) Cascading randomization on baselines generated by a stand-alone GAN lead to little randomization on the saliency maps. (B) Colored-MNIST dataset. GAN baselines generated with both similarity loss and reconstruction loss (S+R), similarity loss only (S), reconstruction loss only (R), and none of those (NA). Only S+R and S successfully constrained the baselines in the same modes (colors) with the inputs.

Table 3: Run Time Analysis comparing the attribution computation time for IG, expected gradient (EG), DeepLIFT (DL) and Occlusion (Occ), as well as the baseline generation time for EG, GANMEX (GAN), and MDTS. The computation was performed on a single Tesla V100 GPU, and the compute time was measure in seconds on calculation over all samples for the dataset, and the baseline generation time is the additional compute time on top of the attribution methods. (§) We selected the same sampling number for IG and EG, and the baseline selection time of EG was estimated by the complexity difference of EG and IG. (†) The attribution inference time for CIFAR10 was measured in 10 separate batches due to the memory constraint. (‡) MDTS search was performed on CPU instead of GPU.

| Dataset | Size | Dim. | Attribution Inference | | | | Baseline Generation | | | GAN Training | |
| | | | IG | EG | DL | Occ | EG§ | GAN | MDTS‡ | Steps | t (hour) |
| --- | --- | --- | --- | --- | --- | --- | --- | --- | --- | --- | --- |
| MNIST | 10k | 784 | 43.5 | 64.1 | 0.6 | 75.7 | 20.6 | 92.5 | 850.8 | 100k | 5.2 |
| SVHN | 26k | 3072 | 557.3 | 658.9 | 1.9 | 4917.0 | 101.6 | 399.8 | 3674.4 | 300k | 18.2 |
| CIFAR10† | 10k | 3072 | 239.4 | 294.8 | 30.0 | 1532.1 | 55.3 | 959.7 | 1085.8 | 400k | 23.8 |

**Classification Loss** ($\lambda_{\text{cls}}^{f}$) Low classification loss tended to make some transformation unsuccessful, and high classification loss introduced additional noise that make the images unrealistic.

**Similarity Loss** ($\lambda_{\text{rec}}$) Similarity loss is the key component for minimum distance optimization. As we have shown in Section D and Figure 7.B, at zero similarity loss, the generator is only constraint by the reconstruction loss and can lead to incorrect font colors and background. High similarity loss, on the other hand, makes the baselines to be too similar to the original images.

**Reconstruction Loss** ($\lambda_{\text{sim}}$) As we have mentioned in Section D and Figure 7.B, reconstruction loss is not required for GANMEX, but it slightly helps GAN to converged. In contrast, high reconstruction loss can lead to incorrect outputs.

## F    COMPUTE TIME ANALYSIS

In Table 3, we measure the GANMEX compute time compared with various attribution methods. While the GAN component takes 5-23 hours to train depending on the datasets, the inference step only requires one single forward operation, and the compute time (MNIST: 9.3 ms, SVHN: 15.4 ms, CIFAR10: 96.0 ms) is at the same order with IG and Occlusion. More experiment details are provided in the caption of Table 3.

## G    INTUITIONS BEHIND THE MINIMUM DISTANCE REQUIREMENTS

Here we present the intuitions behind the baseline selection criteria from Section 3.1 using a simplified formulation. Assuming a transformation $\rho$ projecting from a set of high-level concept variables $V$ to a sample $x$, with $x = \rho(V)$, and we can separate $V$ into three groups $V = \{V^{\text{dis}}, V^{\text{con}}, V^{\text{irr}}\}$. Here, $V^{\text{dis}}$ are the discriminating variables that leads to the model decision, for example, the color the fruits in our apply/orange dataset; $V^{\text{con}}$ are the contingent variables that are independent to the model decision but correlate with how the discriminating variables are presented (eg. size and location of the fruits); $V^{\text{irr}}$ are the irrelevant variables that are both independent to the model decision and uncorrelated to the presentation of the discriminating variables (eg. background colors of the images). The expected one-vs-all explanation under the formulation would be

$$A_{c_o}(x) = A_{c_o}(\rho(V)) = A(\rho(V_{c_o}^{\text{dis}}, V^{\text{con}}, V^{\text{irr}})) = \alpha_{c_o}(V_{c_o}^{\text{dis}}, V^{\text{con}}) \tag{10}$$

with $\alpha$ being a transformation from the underlying concept variables to the explanation. Here we assumed a correct mapping $\alpha$ should be independent of $V^{\text{irr}}$ because the variable set has no impact on the discriminating variables themselves or how discriminating variables are presented.

Now if we apply the concept of one-vs-one, we expect a one-vs-one explanation to produce

$$A_{c_0 \to c_t}(x, B_{c_t}(x)) = \alpha_{c_0 \to c_t}(V_{c_o}^{\text{dis}}, V_{c_t}^{\text{dis}}, V^{\text{con}}) \tag{11}$$

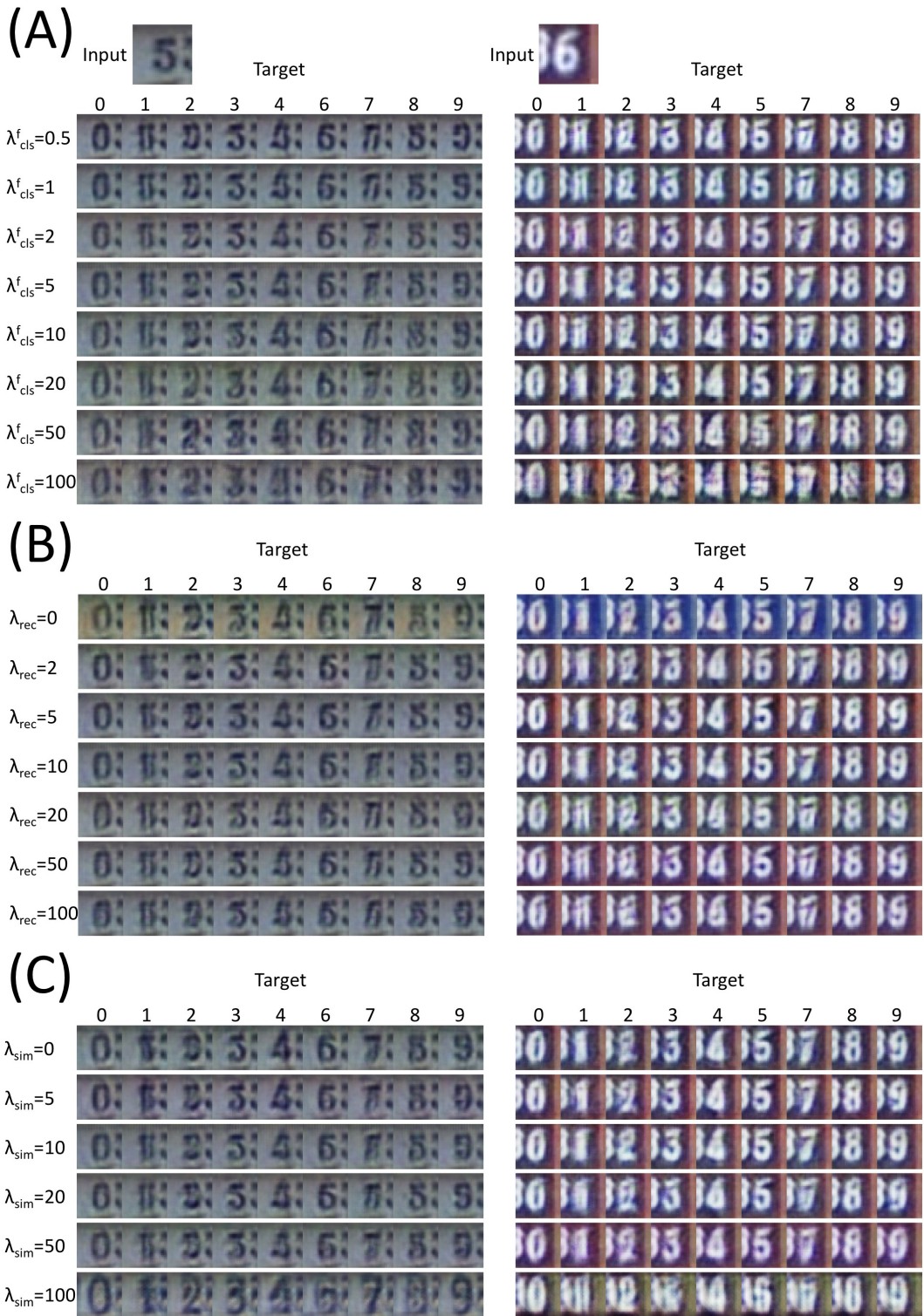

Figure 7: GANMEX baselines generated with various weights for the (A) classification loss, (B) similarity loss, and (C) reconstruction loss.

where $A_{c_0 \to c_t}$ and $B_{c_t}$ were defined in Section 3, and $V_{c_o}^{\text{dis}}$ and $V_{c_t}^{\text{dis}}$ are the discriminating variables for class $c_o$ and $c_t$. In the apple to orange example, $V_{c_o}^{\text{dis}}$ would represent red color, and $V_{c_t}^{\text{dis}}$ would represent orange color.

For a baseline gerenative function $B_{c_t}(x) = B_{c_t}(\rho(V^{\text{dis}}, V^{\text{con}}, V^{\text{irr}})) = \rho(\tilde{V}^{\text{dis}}, \tilde{V}^{\text{con}}, \tilde{V}^{\text{irr}})$, where $\tilde{V}^{\text{dis}}$, $\tilde{V}^{\text{con}}$ and $\tilde{V}^{\text{irr}}$ are the concept variables for the generated baseline input. We can write out

$$A_{c_0 \to c_t}(x, B_{c_t}(x)) = A_{c_0 \to c_t}(\rho(V^{\text{dis}}, V^{\text{con}}, V^{\text{irr}}), \rho(\tilde{V}^{\text{dis}}, \tilde{V}^{\text{con}}, \tilde{V}^{\text{irr}})) \tag{12}$$

Although $A_{c_0 \to c_t}$ can be designed to be independent of $V^{\text{con}}$ and $\tilde{V}^{\text{con}}$, we still need additional constraint to make Equation 12 satisfy the form of Equation 11. While one can apply additional constraints on $A_{c_0 \to c_t}$ to satisfy such requirements, an alternative approach would be requiring $B_{c_t}$ to ensure the following.

$$\tilde{V}^{\text{dis}} = V_{c_t}^{\text{dis}} \tag{13}$$
$$\tilde{V}^{\text{con}} = V^{\text{con}} \tag{14}$$
$$\tilde{V}^{\text{irr}} = V^{\text{irr}} \tag{15}$$

Equation 13 requires the baseline to belong to the target class $c_t$, and this implies that without additional constraints to $A_{c_0 \to c_t}$, a class-targeted one-vs-one baseline is required for correct one-vs-one explanations.

Equation 14 and Equation 15 combined have led to the closest input requirements in Section 3.1. Assuming a smooth transformation ($\rho$), minimizing the distance of $\|x - B_{c_t}(x)\|$ provides an effective way of ensuring Equation 14 and Equation 15. Going back to the apple/orange example, a baseline satisfying Equation 13-15 for an apple image input would be an image with an orange fruit of the same size, at the same location, with the same background to the original input, and all of the above can be achieved by the minimum distance sample described in Section 3.1.

## H  ADDITIONAL FIGURES

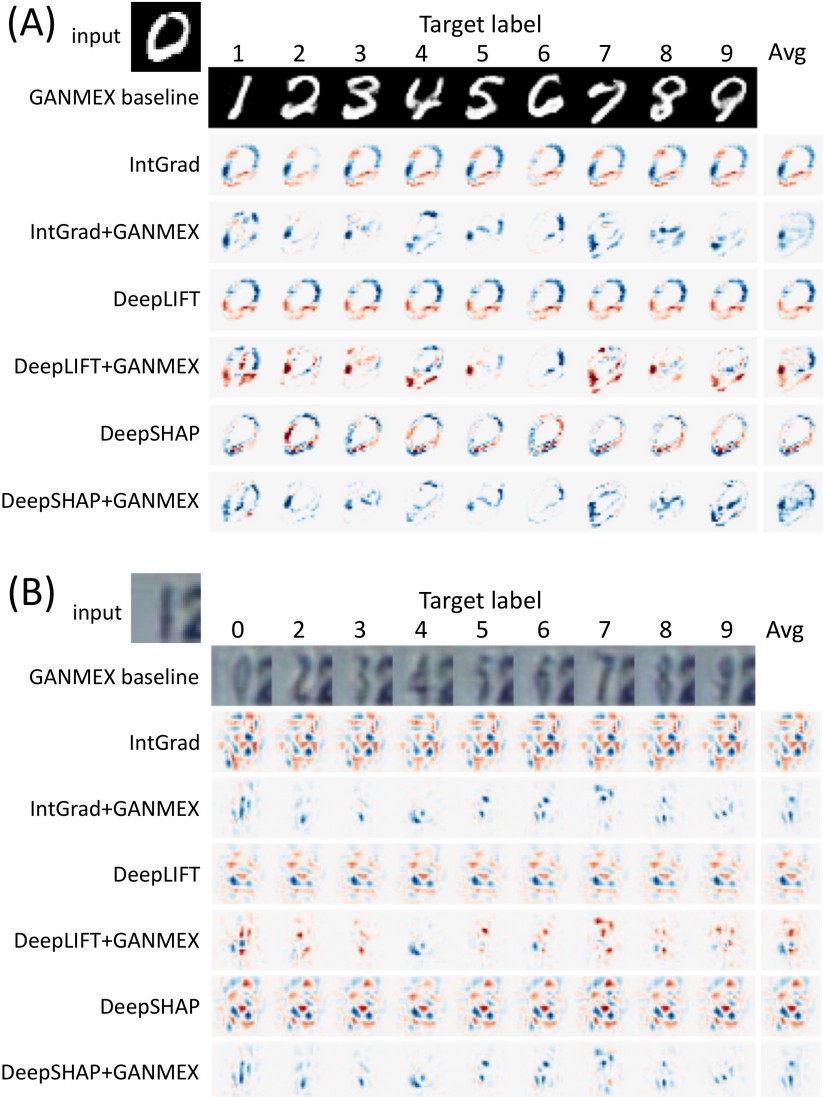

Figure 8: One-vs-one saliency maps using class-targeted baselines (GANMEX) vs non-class-targeted baselines (zero baselines). One-vs-one saliency maps generated using zero baselines show almost the same attributions regardless of the target class, making the one-vs-one saliency maps (columns with target labels) similar to the one-vs-all saliency maps (the "Avg" columns that show the averaged saliency maps over all target classes). GANMEX baselines corrected the behavior for both IG, DeepLIFT and DeepSHAP, and produced different attributions depending on the target classes.

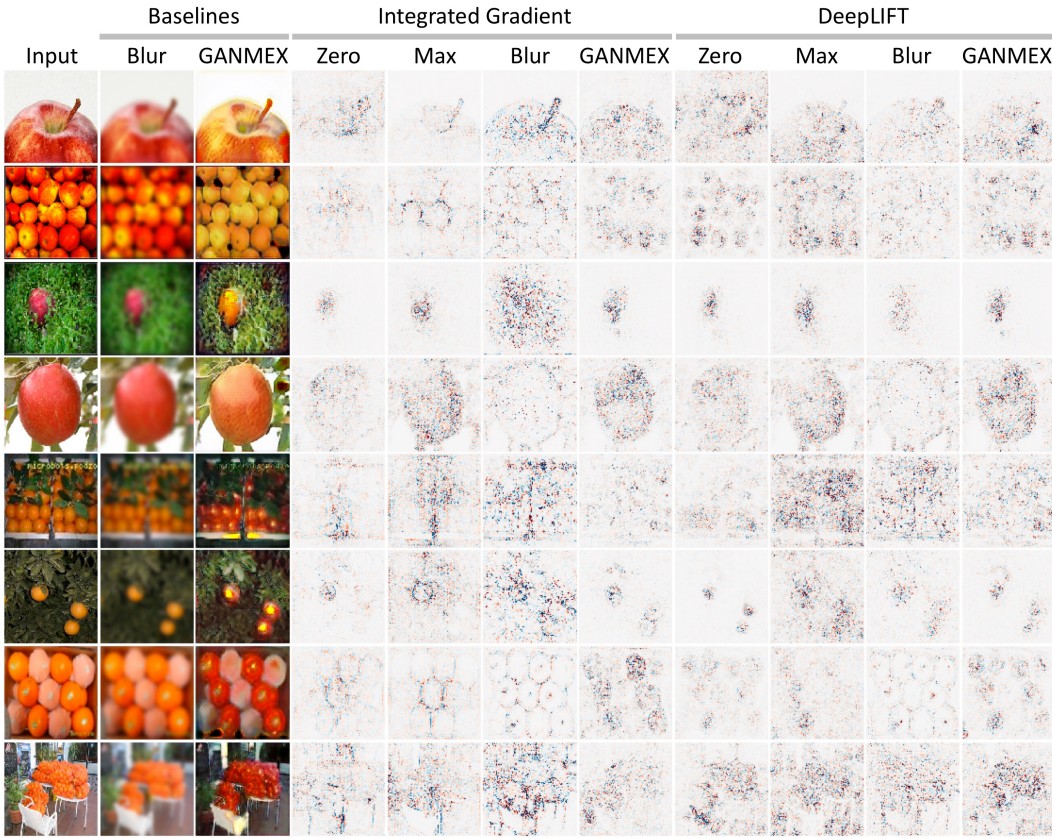

Figure 9: Additional examples of saliency maps for the classifier on the apple2orange dataset with four baseline choices: zero baseline (Zero), maximum value baseline (Max), blurred baseline (Blur), and GANMEX baseline (GANMEX).

