# OpenReview forum: "GANMEX: Class-Targeted One-vs-One Attributions using GAN-based Model Explainability"
_ICLR.cc/2021/Conference — Reject_

### Official Review · AnonReviewer3 · 2020-10-26
**Recommendation to Reject**

**Rating:** 4
**Confidence:** 2

**Review:**

The paper claims to present a novel GAN-based model explainability, for generating one-vs-one explanations, by incorporating to-be-explained classifier as part of the GAN. They use GANs to produce a baseline image which is a realistic instance from a target class that resembles the original instance.

Positive aspects:
- a novel approach for generating one-vs-one explanation baselines leveraging GANs
- the proposed approach improves the saliency maps for binary classifiers

Negative aspects:
- the paper lacks clarity
- the approach is demonstrated on cherry-picking examples. Have doubts of its generalization capability

Please find below some of my concerns:
1. Your claim: "...we use GANs to produce a baseline image which is a realistic instance from a target class that resembles the original instance". Why do you need a GAN? Why don't you use a network to generate a confusion matrix to analyze the performance of the classifier? And based on this analysis you could explain why, for instance, the digit '0' is classified as a '6'.
2. Related with the previous point, your analysis is very limited. You assume '0' is classified as '6'. Could '0' be classified as an '8' or '9'? It is not clear from your analysis. There are no comments on these cases. Looks like your examples to defend your approach are cherry-picked.
3. I am not sure how to interpret Figure 2.
Some other comments:
1. The paper lacks novelty. The authors' contribution is not clear.
2. The experimental validation is limited and not convincing. The authors use just some simple datasets (MNIST, SVHN). What about more complex datasets, like CIFAR10, LSUN, etc.? Could your approach explain the mis-classification in these cases?

---

> ### Author Response · Authors · 2020-11-24
> **Response to Review #3**
>
> We are grateful for the reviewer’s constructive feedback and inputs. We have added a mis-classification analysis to Section 4.1 which would further adds value to our work. However, we respectfully disagree with the reviewer’s assessment about our work. The reason that a limited number of examples was shown in the main section of our manuscript was because of the space constraint; however, we have included additional examples in the Appendix sections. Additionally, as suggested by other reviewers, we have included multiple metrics in Appendix C and Table 2, in addition to the evaluation approaches in Section 4.1 that we originally presented, and the reported metrics have pointed to the strong performance of our approach.
>
> Please find our reply to the reviewer’s comment below:
>
> 1. Similar to other attribution methods [https://arxiv.org/abs/1703.01365, https://arxiv.org/abs/1704.02685, https://arxiv.org/abs/1711.06104], our one-vs-one attribution approach provides instance-wise explanations, meaning that the method would provide explanations for a trained model decision on a particular sample. Take the model trouble-shooting use case for example, the question one-vs-one attributions try to answer is "why the model classifier *this image* as a '0' and not a '6', and we expect a successful attribution output to highlight the features on the particular image that have lead to the model decision. To the best of our knowledge, a confusion matrix would not provide such insights.
>
> 2. We would like to point the reviewer to Figure 8 of our manuscript (was included as Figure 6 in the originally submitted version) where we have provided the generated baseline and saliency maps for all target classes. We particularly picked 0 -> 6 to demonstrate in the main figure, because ‘0’ and ‘6’ are relatively similar digits, so it’s easier for the readers to judge the correctness of the saliency maps. The rationale behind the other choices of (original class, target class) pairs were the same.
>
> 3. Figure 2 (now Figure 2.A) is in fact the primary results of our work. The first column shows the input samples, and the 2nd-3rd column shows the baseline image selected by the in-sample search (MDTS) and our approach (GANMEX). We would expect good baseline images to be “close” to the original images, and this has been achieved better by GANMEX. The rest of the columns shows the saliency maps generated with different attribution methods (integrated gradient, expected gradient, DeepLIFT, occlusion, and DeepSHAP) with different baseline image choices (zero input, MDTS, GANMEX). Successful one-vs-one saliency maps should highlight the key features that lead the model to decide on the predicted class $c_o$ instead of another specified class $c_t$. Based on the results, we can visually assess that GANMEX leads to better explanation compared with other methods.
>
> 4. We would like to reiterate that the contributions of our paper are the followings:
> (1) We developed a novel design applying GAN for the one-vs-one attribution problem
> (2) We performed detailed analysis and evaluation comparing different baseline choices, and showed that GANMEX outperformed others baselines.
> (3) We addressed the problem of existing attribution methods failing the sanity check suggested by [https://arxiv.org/abs/1810.03292], which has been a major challenge to the field.
>
> 5. We thank the reviewer for the valuable feedback. We have added the CIFAR10 data the mis-classification analysis to our manuscript.

---

### Official Review · AnonReviewer4 · 2020-10-28
**Official Blind Review #4**

**Rating:** 5
**Confidence:** 4

**Review:**

Paper Summary:

This paper considers the less-explored baseline selection issue in attribution methods for one-vs-one explanations of multi-class classifiers. The key insight is to construct the closest and realistic target class baseline. To this end, an existing image-to-image translation GAN model, namely StarGAN, is leveraged to transform an input example to another example in a target class yet is close to the input. This baseline can be integrated with a variety of attribution methods, including integrated gradient, DeepLIFT, Occlusion, and deepSHAP, and shows consistent improvements over zero baseline and minimum distance training sample for one-vs-one explanations. The experiments are conducted on three datasets – MNIST, SVHN, and apple2orange.

Paper Strengths:

This paper addresses an important yet overlooked baseline selection problem. The way the authors address this problem is interesting by leveraging GAN models. Empirical evaluations demonstrate the effectiveness and generalizability of the proposed approach.

Paper Weaknesses:

1) The main weakness of this paper to me is the evaluation section. The proposed approach is only validated on simple datasets like MNIST and SVHN. It would be more convincing to show the effectiveness of the proposed approach on natural images and a large number of classes, like CIFAR and ImageNet, as used in the previous work such as IG.

2) Following the comment in 1), the prior works, such as IG and DeepLIFT, have been used to analyze other types of models and been evaluated on other types of data, such as genomics and neural machine translation. In addition to images, would the proposed approach also apply to these domains?

3) As illustrated in Figure 1, the key assumption of the proposed approach is that a GAN model (StarGAN) is able to generate examples that are much closer to the input examples than the training examples (i.e., minimum distance training sample). Under what conditions would such an assumption hold?

4) Following the comment in 3), it would be interesting to show and analyze some failure cases.

5) In the proposed approach, the StarGAN directly uses the already trained model classifier as its discriminator. How if the StarGAN trains its own discriminator without using the model classifier?

6) It would be interesting to show the hyper-parameter (different trade-off lambdas) sensitivity.

7) I understood that the authors focused on one-vs-one explanations. But I am interested to hear the authors’ thoughts on how to extend the proposed approach to one-vs-all explanations.

After Rebuttal:

I thank the authors for the rebuttal. I have also read the other reviewers’ comments. Unfortunately, the rebuttal is unconvincing and sometimes vague. I keep my original rating.

---

> ### Author Response · Authors · 2020-11-24
> **Response to Review #4**
>
> We sincerely thank you for the thorough review and feedback. Please find our reply below.
>
> 1. To further justify that our approach works well for natural images, we have added the results for CIFAR10 to our manuscript (Figure 2,3). The existing results on the apple/orange also represent natural images (Figure 4,9). Large number of classes is currently the limitation to our methods with our computational resource, and we have mentioned this limitation in the future work section of our paper.
>
> 2. We appreciate the comment regarding non-image datasets, but the priority of focus for this manuscript was on saliency maps for images. Saliency maps - the image form of attributions has formed a challenging area with a wide range of unanswered questions. Hence, to our knowledge, the current study is the first to address the issues raised by "Sanity Checks for Saliency Maps" [1] using alternative baseline choices. We feel like our research on the saliency maps along has contributed greatly for bridging the research gap.
>
> 3. Thank you for the valuable suggestions on further analysis around sample distance minimization. Under our baseline definition, baseline search is a constrained optimization problem, constrained by the set of “realistic samples belong to the target class”. The MDTS approach searches among the training dataset, which is finite subset within the overall search space, and therefore MDTS can lead to suboptimal results compared to GAN-based optimization. The effectiveness of sample search method depends on (1) the dimension of the parameter space, (2) the dataset diversity, and (3) the number of training samples. We have added Appendix B with Figure 5 and Table 1 for analyzing the similarity between sample and baseline inputs. We showed that for high diversity dataset (measured by the average inter-class distance) such as SVHN and CIFAR10, GAN-based baseline generation has significantly outperformed the sample search approach (MDTS).
>
> 4. Following #3, we have added the failure case analysis in Appendix B and Figure 5 of our manuscript.
>
> 5. We would like to bring the reviewer's attention to Appendix C and Figure 6.A in the revised version (was Appendix B and Figure 5.A in the previous version), where we have reported this experiment. When the StarGAN trains its own discriminator, the explanation will be partially based on the StarGAN’s discriminator in additional to the to-be-explained discriminator. We have tested the explanation correctness using the sanity check proposed by [1] and showed that StarGAN’s own discriminator has led to undesirable outcome.
>
> 6. Thank you for the suggestion. We have added the hyper-parameter analysis in Appendix E and Figure 7.
>
> 7. One-vs-all explanation is a highly interesting area for our future work. One approach we would potentially like to explore is to calculate the ensemble of saliency maps across all target class $A_{S,c_o}^{1-vs-all} = E_{c_t} A^{1-vs-1}_{S,c_o \to c_t} (x, B_{c_t}(x)) = \sum_{c_t} A_{S,c_o \to c_t} (x, B_{c_t}(x)) P(c_t|x)$,
>
> where $A^{1-vs-1}$ is the one-vs-one saliency defined in our paper, $E_{c_t}.$ is the expectation over all target classes $c_t$, and $P(c_t|x)$ is the probability conditioned on the sample. The ensemble concept is similar to smoothgrad and expected gradient but the baselines will instead be generated by GAN and specific to the target classes. If we adopt the blank assumption of uniform $P(c_t|x)$, then $A^{1-vs-all}$ becomes a simple average of $A^{1-vs-1}_{S,c_o \to c_t}$ over all $c_t$, which was shown in the "Avg" columns of Figure 8.
>
> [1] https://arxiv.org/abs/1810.03292

---

### Official Review · AnonReviewer2 · 2020-10-28
**Using GANs to get baselines when finding feature attributions**

**Rating:** 5
**Confidence:** 3

**Review:**

Summary: This paper looks to use GANs to generate baselines for attribution methods. The focus on one-vs-one feature importance explanations is novel, to the best of my knowledge. The paper attempts to make progress on the baseline selection problem that has plagued the feature importance community.

Strengths
- As far as I know, the authors' contribution of one vs one attribution (compared to one vs any attribution) is novel. Whilst other works have alluded to this or ran heusristic experiments, this paper does a good job of formalizing the notion.
- The ability for GANMEX to live on top of any other attribution method makes it an attractive addition to existing attribution methods. Thank you for visualizing the baselines generated by GANMEX, quite helpful :)

Weaknesses
-  A computational complexity analysis is required to gauge the practical utility of generating baselines with GANMEX. Also, it would be nice to give complexity of GANMEX compared to FIDO, EG, and simple nearest neighbor baselines.
- In addition to the visual comparisons provided, it would have been helpful to evaluate explanations using exisiting evaluation criteria in the attribution literature (i.e., faithfulness, sensitivity, monotonicity, etc.) This paper has the opportunity to broadly assess the effects of various baselines on attributions.

Questions
- While GANs seem like an attractive choice of deep generative model (DGM) for this problem, can you comment on or experiment with other DGMs (i.e., VAEs or specifically VAEACs [1])? However, any DGM that has latent class separation should suffice. You would be able to perform optimization in the latent space [2, 3, 4] and achieve similar class separation, as described in Figure 1.
- The attributions in Figure 3.E seem like noise, while zero baseline seems visually appealing -- can you provide some intuition for why this occurs? The GAN feels like overkill for MNIST, but might be suitable for other high dimensional problems wherein the baseline needs to pick up on small nuances in the data.

[1] https://openreview.net/forum?id=SyxtJh0qYm

[2] https://arxiv.org/abs/1806.08867

[3] https://arxiv.org/abs/2006.06848

[4] https://arxiv.org/abs/1807.08024

---

> ### Author Response · Authors · 2020-11-24
> **Response to Review #2**
>
> We would like to thank you for recognizing the novelty of our work and providing the constructive feedbacks. We will address the reviewer’s concerns and questions below:
>
> **Computational Complexity**
>
> We've added a complexity analysis in Table 3 comparing GANMEX with EG and simple nearest neighbor baselines as well as the associated attribution methods. FIDO is one-vs-all instead of one-vs-one, so we excluded it in the current draft for the interested of completing all the revisions, but we would like to include FIDO runtime analysis in the next version of the draft.
>
> GANMEX does requires a one time GAN training process, similar the FIDO/CA-GAN combination, the reported state-of-the-art for one-vs-all saliency maps. However, the inference time is feasible for most practical purposes as it was comparable to the attribution methods (IG, EG, occlusion) themselves.
>
> **Evaluation**
>
> Thanks for the valuable suggestions. We have added additional metrics including AOPC, faithfulness, monotonicity, and inverse localization to Appendix C and Table 2 of our manuscript.
>
> The sensitivity aspect of saliency maps was measured via multiple metrics described above. The single-pixel sensitivity was implicitly measured by the faithfulness metric. Sensitivity-n proposed by [https://arxiv.org/abs/1711.06104] was covered by AOPC, which calculates the cumulative sum over the top-L features. Inverse localization measures the “insensitivity” concept by encouraging the saliency maps to be less sensitive in the non-focus area.
>
> **Question about DGM choices**
>
> Based on what was presented the literature [5], VAEs tend to generate overly smooth or blurry images. [2] and [3] are both potential baseline selection methods for saliency maps, but based on the results provided, the examplar and VAE based approaches both led to blurry images. While deliberately blurred baselines have been used for attribution methods in the past, they have led the attribution results to overly focused on the edges or high-frequency features, as shown in [6] and Figure 4 in our manuscript.
>
> While GANs generation requires higher training complexity compared to other DGMs, it was a price to pay for preserving the sharpness of images and lead to improvements in the saliency maps. The inference time was in the acceptable range - comparable with the attribution methods themselves.
>
> FIDO [4] represents a separate line of research area for one-vs-all explanation. We have added a paragraph in our manuscript comparing between FIDO and our approach. VAEAC [1] requires partial images as conditioning, so it would work for one-vs-all explanation using frameworks like FIDO, but not one-vs-one explanation.
>
> **Noise Images from Sanity Checks**
>
> Figure 3.E demonstrated the result of sanity checks proposes by Adebayo et al [7]. The experiment was performed by randomly perturbing the model parameters in a cascading manner. The expected outcome is to produce randomized attribution after the model parameters being randomized. Therefore, less appealing attributions under the model cascading perturbation is in fact more desirable.
>
> The undesirable checks result of saliency maps being insensitive to model randomization has been a known problem for the major attribution methods/saliency maps. One major contribution of our work is in fact "fixing" the fore-mentioned problem in the existing attribution methods by swapping the default baselines with GAN-generated baselines and making them produce the expected sanity check outcome.
>
> [1] https://openreview.net/forum?id=SyxtJh0qYm
>
> [2] https://arxiv.org/abs/1806.08867
>
> [3] https://arxiv.org/abs/2006.06848
>
> [4] https://arxiv.org/abs/1807.08024
>
> [5] https://arxiv.org/abs/1607.07539v3
>
> [6] https://distill.pub/2020/attribution-baselines
>
> [7] https://arxiv.org/abs/1810.03292

---

### Official Review · AnonReviewer1 · 2020-10-29
**Considers an important problem but conclusions not clear**

**Rating:** 5
**Confidence:** 4

**Review:**

Summary\
This paper proposes a new 'baseline' for attribution methods tailored to deep neural networks. DNN attribution methods like integrated gradients, deep lift and others require a baseline to compare to as part of the computation. The choice of a baseline has been controversial in the literature, and a good method to select a baseline remains an open problem. This paper seeks to address that problem. Specifically, this paper seeks to develop a baseline for one-vs-one explanations as opposed to one-vs-all explanations. Consider an MNIST model, a one-vs-one attribution would attribute why an input is say a '2' and not a '4', i.e., it is contrastive against a particular target class and not all classes. This paper proposes to use a StarGAN for generating these baselines. The paper then evaluates explanations derived using the new baseline and shows that they explanations 'perform' better.

Overall, I think the paper tackles an important problem, but I have several concerns with the motivation, the appropriateness of the baseline definition in this work, and the evaluation. I'll expand on these concerns in the later part of the review, so I am not recommending an accept in its current form.

Significance/Quality\
The paper tackles an interesting and potentially challenging problem. However, motivation is still somewhat unclear, and there are critical problems with the evaluations used as justification here. I go into these at the end of this review.

Clarity/Writing\
The paper is generally easy to follow. I problem I had reading it is that there are a few sentences that are stated as fact without any justification. For example, the paper notes, "The minimum distance training sample in section 2.2 is a true class-targeted baseline proposed in the past." What is a 'true' class-targeted baseline? Such statements should probably be reformulated.

Minor changes
Lasted paragraph of section 2.3: "has posted a challenging", posted is probably not the desired word here.

Questions and Concerns

- Motivation: it is still not clear to me why a one-vs-one attribution is desirable? More should probably be done here to motivate this. The biggest need though is motivation for the one-vs-one attribution baseline. In several statements, the paper alludes to properties of baselines used in expected gradients and other methods, stating the reasons why these baselines are undesirable. I agree, but why should a one-vs-one baseline be preferable to these? Ideally, the paper will set out a list of desirable properties; then show that the baseline derived from GANMEX satisfies these. It is still not clear to me why a notion of minimum distance in a different target class is the right one. Can the authors say more about why this should be the case?

- Evaluation: I'll preface my concerns here with the fact that I think evaluating model attributions or explanations in general is a difficult and open problem. This said, I don't think any of the evaluations presented in this paper can be taken as showing that the GANMEX baseline is the desirable one. First, the perturbation-based evaluation does not provide consistent rankings (see: Tomsett et. al. (AAAI Sanity checks for saliency metrics)). I suspect the gini index will have the same problems as those discussed in the Tomsett et. al. paper. The sanity checks themselves, i.e. the cascading randomization, will tell you if a method should be ruled out and not whether a method is effective. Consequently, I don't think the sanity checks can say much in judging baselines. Having said all of this, I think the way to evaluate a baseline is to take a task where the truth ground-truth rankings are known a priori, train a model to respect and align to the true ground truth. Now one can compare attributions from such a model for a normal baseline and a baseline from GANMEX. Assuming the attribution method itself is a reliable one, then one can quantify improvements due to the GANMEX baseline.

A paper that might be related to this work that also incorporated generative modeling: https://arxiv.org/abs/1807.08024.pdf

Overall, the concerns above make me hesitant about this current draft; however, I am happy to revise my assessment if the authors think I am wrong.

---

> ### Author Response · Authors · 2020-11-24
> **Response to Review #1**
>
> We sincerely thank you for providing an insightful review. We would like to address the reviewer's concerns in the followings.
>
> **Motivation**
>
> We understand reviewer’s concern, however, the underlying reason that one-vs-one explanations are less commonly adopted are that their methodologies were not as well-developed compared to one-vs-all explanation.
>
> For multi-class classification, one-vs-one questions ("why this and not that") are natural questions to ask, and they are complementary to one-vs-all questions. To fully understand the model's decision, it often comes down to how the model distinguishes between similar classes. In some practical cases, we would argue that one-vs-one explanations can provide more valuable insights than one-vs-all explanations. For example, in medical diagnosis, why a patient was diagnosed with disease A but  not disease B, or in biological taxonomy studies, why an organism belongs to species C but  not species D. For diagnosing model performance, one-vs-one explanations provide insights into why an instance was not classified into the correct class. We have added an analysis under Section 4.1 in our manuscript to further demonstrate how one-vs-one saliency can be applied for explaining mis-classification examples.
>
> One-vs-one explanation is an existing but unsolved problem. In the original work of DeepLIFT and DeepSHAP, the authors have provided one-vs-one explanations via the definition described in Section 2.3. However, without appropriate choices of one-vs-one baselines, the results were generally unsatisfying. While conceptually we would expect desirable one-vs-one to be dependent of the target classes, we have showed empirically (Figure 8) that one-vs-one saliency *without* one-vs-one baselines are only marginally independent of the target classes, and such outcome was undesirable even based on simple visual assessments.
>
> In addition to the experimental results, we have added Appendix G providing intuition behind (1) why a one-vs-one baseline is required for one-vs-one explanations and (2) why a minimum distance between the sample and the baseline input is preferable. At the high level, the attribution methods used the baselines as guidance for feature perturbation, and it's unlikely to produce the correct attribution if the object in the original input and baseline input are located differently. Therefore, a good baseline generation method would preserve the contingent variables (location, size of the object, ...), and that's what the minimum distance constraint tries to achieve.

---

> > ### Author Response · Authors · 2020-11-25
> > **Response to Review #1 (continued)**
> >
> > **Evaluation**
> >
> > The saliency evaluation remains to be a very active field of research, and there is still inconsistency between the proposed metrics. Despite all these, one of the most significant contributions of our work regarding performance improvement was addressing the cascading randomization sanity checks. The checks proposed by Adebayo et. al. tested relatively simple properties of saliency maps - the explanation should be sensitive to the model parameters. The tested concept was widely accepted by the field, and yet the existing attribution methods failed this simple validation. In our work, we showed that such properties can be achieved by imposing "smarter" baselines to the existing attribution methods, instead of having to develop new attribution methods.
> >
> > Since the sanity check provide (roughly) binary outcomes, it would not have said much if all methods pass the judgement. In our case, it was known that the existing saliency maps generate by IG or DeepLIFT failed the checks, and the fact that the GANMEX-modified saliency maps pass the sanity checks has made a strong distinction between our approach and the existing methods.
> >
> > We agree that a ground-truth dataset would add credibility to our evaluation analysis. Constrained by the lack of existing dataset with annotated ground-truth explanations (at least to the best of our knowledge), we used SVHN as a proxy, in which the vertical edges areas of the images generally do not contain critical information (>95% of the time based on our observations). We then tested whether the explanations were constrained to the area of interest by calculating the "inverse localization", which measures the relative intensity between the outside area and the focus area. The results showed that the baseline choices closer to the input images (GANMEX and MDTS) have led to significantly more focused saliency maps compared with zero baselines and random baselines (expected gradient). Furthermore, the GANMEX variant performed the best for all attribution methods tested.
> >
> > Regarding the evaluation metrics, Gini index measures the saliency map sparsity, which is a desirable property for one-vs-one saliency maps to have, because one would expect one-vs-one explanations to highlight less features compared to one-vs-all explanations. We have added more measures and dataset suggested by other reviewers, and we did find the inconsistency as reported by Tomsett et. al. Our interpretation (aligned with Tomsett et. al.) is that not all saliency metrics measure the same underlying concepts. In particular, faithfulness and monotonicity, measure the impact of changing single pixels, which is more in line with the adversarial robustness measures.
> >
> > We have addressed the clarity/writing related comments raised by the reviewer. Also thanks for pointing out the FIDO paper. We have added a paragraph comparing FIDO with our method.

---

### Author Response · Authors · 2020-11-25
**Revision Summary**

We would like to thank all the reviewers for the insightful comments which greatly improved the quality of the manuscript. We are grateful for the valuable feedback and have included the following revisions according to the reviews.

1. We included the CIFAR-10 dataset into our examples and evaluation results (Table 2, Figure 2-3).
2. To evaluation the saliency maps with different baseline choices, we designed a pseudo ground-truth evaluation method with an inverse localization metrics, and the results showed that GANMEX consistently outperform other baseline choices. (Table 2, Appendix C).
3. We added additional metrics for assessing the different baseline choices, including $AOPC_L$, $AOPC_{all}$, Faithfulness, and Monotonicity (Table 2, Appendix C).
4. A compute time analysis comparing GAN with other baseline selection approaches was presented in Table 3 and Appendix F.
5. We added a detail analysis on the instance to baseline distance for evaluating the effectiveness of various baseline selection approaches (Appendix B).
6. A hyper-parameter analysis was added as Appendix E.
7. We presented examples where one-vs-one saliency maps help explains the mis-classification  (Figure 2.B, Section 4.1)
8. We included a formulation that leads to the intuitions behind two of our fundamental assumptions: 1) a class-targeted baseline is required for one-vs-one attributions, and 2) a baseline with the closest distance to the sample is preferable (Appendix G)
9. A paragraph was added to Section 2.4 for comparing our approach with FIDO, the current state-of-the-art for GAN-based one-vs-all explanations.

Thanks again for your time and please let us know if there is any additional feedback we could address.

---

### Decision · Program_Chairs · 2021-01-07
**Final Decision**

**Decision:**

Reject

**Comment:**

This work investigates the choice of a 'baseline' for attribution methods. Such a choice is important and can heavily influence the outcome of any analysis that involves attribution methods. The work proposes doing (1) one-vs-one attribution in a sort of contrastive fashion (2) generating baselines using StarGAN.

The reviewers have brought out a number of valid concerns about this work:

1. One-vs-one attribution appears to be novel, and distinctive enough from the more prevalent "one-vs-all" formulations. I am perhaps more optimistic than the reviewers that such a formulation is in fact useful, but I can see where the hesitancy can come from.
2. It's not clear that the evaluation shows that the proposed method is in fact superior to the others. All the reviewers touched upon this one way or another.
3. Somewhat simplistic datasets used for evaluation (noted that there are CIFAR10 results in the rebuttal).

This was more borderline than the scores would indicate. I thank the authors for the extensive replies and extra experiments. I encourage them to incorporate more of the feedback and resubmit to the next suitable conference. I do believe that doing experiments on ImagetNet (like previous work does, such as IG) would be quite worthwhile and convincing. I suspect the computational expense could be mitigated by re-using pretrained networks, of which there are many available for ImageNet specifically.